**Subject Category:**
Biology (whole organism)

ecology/environmental science/biocomplexity

aggregation, coral – algae, time scales, transient dynamics, pattern formation, state space

**Author for correspondence:**
Marlene Brito-Millán
e-mail: mbritomillan@gmail.com

# Influence of aggregation on benthic coral reef spatio-temporal dynamics

Marlene Brito-Millán[1,2,3], B. T. Werner[1],
Stuart A. Sandin[2] and Dylan E. McNamara[4]

[1]Complex Systems Laboratory, Climate, Atmospheric Sciences, and Physical Oceanography, and University of California - San Diego, 9500 Gilman Drive, La Jolla, CA 92093-0230, USA
[2]Center for Marine Biodiversity and Conservation, Scripps Institution of Oceanography,
[3]Environmental and Ocean Sciences Department, University of San Diego, 5998 Alcalá Park, San Diego, CA 92110-2492, USA
[4]Department of Physics and Physical Oceanography/Center for Marine Sciences, University of North Carolina, Wilmington, 601 South College Road, Wilmington, NC 28403, USA

MB-M, 0000-0002-4265-1528; DEM, 0000-0001-8752-1586

Spatial patterning of coral reef sessile benthic organisms can constrain competitive and demographic rates, with implications for dynamics over a range of time scales. However, techniques for quantifying and analysing reefscape behaviour, particularly at short to intermediate time scales (weeks to decades), are lacking. An analysis of the dynamics of coral reefscapes simulated with a lattice model shows consistent trends that can be categorized into four stages: a repelling stage that moves rapidly away from an unstable initial condition, a transient stage where spatial rearrangements bring key competitors into contact, an attracting stage where the reefscape decays to a steady-state attractor, and an attractor stage. The transient stage exhibits nonlinear dynamics, whereas the other stages are linear. The relative durations of the stages are affected by the initial spatial configuration as characterized by coral aggregation—a measure of spatial clumpiness, which together with coral and macroalgae fractional cover, more completely describe modelled reefscape dynamics. Incorporating diffusional processes results in aggregated patterns persisting in the attractor. Our quantitative characterization of reefscape dynamics has possible applications to other spatio-temporal systems and implications for reef restoration: high initial aggregation patterns slow losses in herbivore-limited systems and low initial aggregation configurations accelerate growth in herbivore-dominated systems.

# 1. Introduction

Coral reefs, one of the most biodiverse ecosystems on the planet, are facing unprecedented losses and degradation in system

function [1]. Although reef decline and degradation have been well documented, the dynamics of reef recovery have received far less attention owing in part to the relatively slow pace of reef growth [2–4]. Characterization of reef recovery dynamics is especially relevant for expensive reef restoration programmes that seek to maximize growth of coral fragments to rehabilitate spaces that were once resilient coral-dominated reefs [5]. Because of the slowly evolving nature of coral reef benthic seascapes (or reefscapes), computer models and complementary long-term monitoring are essential for inferring the mechanisms underlying transient coral reef recovery pathways, with the goal of facilitating possible interventions that minimize the time required to achieve habitat restoration.

The specific mechanisms operating during transient pathways connecting healthy reef states, dominated by reef-building coral and degraded reef states, dominated by fleshy algae, have only recently begun to be investigated as reefscape modelling and field studies increasingly shift their focus from steady-state 'endpoint' or attractor perspectives towards transient dynamics [6–12]. This change in perspective reflects a growing awareness that the time span over which data are collected for scientific and management studies is shorter in duration (months to years) than the time needed for disturbed coral reef systems to decay to an attractor (i.e. a coral-dominated or fleshy algae-dominated end state). Processes (e.g. recolonization or degradation) observed over these short durations probably reflect transient dynamics rather than stability at an attractor.

The dynamics of transient stages in substrate-bound systems such as coral reefs are dependent on interactions between major competitors and are expected to be affected significantly by reefscape spatial patterns [13–19]. Spatial location of individuals strongly influences outcomes of ecological interactions because these interactions occur over relatively short distances and are highly sensitive to the identity of neighbours [20,21]. For example, in non-coral ecosystems such as marsh tussocks of *Carex stricta*, ecological interactions resulting in local growth inhibition from shade and radially accumulating wrack (dead plant material) have been found to affect development of regularly spaced patterns at larger scales [22]. For coral reefs, we expect that intermediate-time-scale behaviour will be controlled, in part, by spatial arrangements that influence competition.

Ecological competition is implicitly dependent on context, and as such can influence dynamics. For example, most competitive interactions are density-dependent, with density of competitors being linked nonlinearly to vital rates (e.g. probability of survival or rate of growth) [20]. Among sessile organisms, the effective density of competitors is determined by the spatial 'halo of interaction' for individuals (area within which one organism can interact significantly with a competitor) and the relative spatial positioning of competitors. Therefore, the degree of nonlinearity is linked to spatial configurations of sessile organisms [19,20]. Such linkages between spatial patterns and dynamics have been studied in chemical reaction–diffusion models, where nonlinearity is excited by spatially localized finite amplitude perturbations that give rise to irregular spatio-temporal patterns significantly affecting how a system arranges itself to arrive at a patterned configuration [23]. In simplified ecological systems, increased aggregation has been shown to increase the degree of nonlinearity (e.g. increased spatial proximity between sessile organisms leads to more intense, localized competitive interactions) and to introduce time lags, or delays in expected time to dominance or extinction of particular organisms [24]; in specialized cases, species coexistence occurs despite competitive asymmetry that would lead to serial extinction in a less-clumped system [18]. Accounting for aggregation patterns when initializing more complicated, competition-based ecological systems is expected to result in time-evolution pathways that are markedly different from those that originate with either a homogeneous or random configuration.

Aggregation is a measure of the degree to which individuals of the same type are spatially clumped, and high levels of aggregation are a common feature in the distribution of substrate-bound organisms, including stony corals [25–28]. Given that the 'halo of interaction' for most corals is limited in extent, often constrained largely to immediate neighbours with shared borders, conspecific aggregation is expected to have a direct impact on realized competition. By changing the length of borders open to interactions between competitive, heterospecific groups, higher levels of aggregation result in individuals interacting less with competitors in other groups and more with members from the same group than would be expected from tracking overall abundance [14]. For example, in coral and ascidian aggregation experiments, the rate with which strong competitors take over space is significantly reduced when aggregated, as their resources are redirected towards competing with each other, allowing weaker competitors to persist longer [15,17]. In reef restoration experiments focused on individual coral colony health, close spacing between coral fragment outplants has been found to increase branching coral vertical growth rates, although overall fitness and long-term survival decreases [29]. In simulated agent-based coral fragment models, growth is maximized in uniform, evenly spaced, gridded coral transplant arrangements, although neither competition with algae nor herbivory level were considered [16]. Patterns of

aggregation are likely to influence the dynamics of coral reef benthic organisms at spatial scales larger than that of the single individual, extending to reefscape patterns and their longer time scales.

We used a lattice model of interacting coral and algae organisms parametrized for Caribbean coral reefs to investigate the dynamical evolution of hierarchically ordered (transitive) competing functional benthic groups with varying initial aggregation. Because coral reefs are complex systems, we employed the theoretical and analytical methods of complexity to investigate the following questions:

(1) How do patterns of initial aggregation affect coral reef system pathways towards endpoint attractors and how do differences depend on herbivore abundance (i.e. reef conservation scenarios)?
(2) Do pathways exhibit linear (one-way) or nonlinear (strong two-way) dynamics and what can be inferred from the presence or absence of nonlinearity?
(3) How do biologically driven diffusive processes (∼colony fusion) contribute to the development of aggregated reefscape patterns?
(4) What minimum set of variables (state space axes) characterize reefscape transient dynamics and evolution towards the attractor?

## 2. Methods

### 2.1. Spatially explicit reef benthic model

Simulations were undertaken using a spatially explicit lattice model that previously has been used to investigate coral reef benthic behaviour under spatially varying patterns of herbivory [7,30]. The model domain consists of a $200 \times 200$ cell lattice simulating a $400 \, \text{m}^2$ area with periodic boundary conditions in both horizontal directions (i.e. a toroidal lattice); large enough to capture cross-reef spatial patterns based on constituent-scale dynamics (for thousands of coral colonies). Additionally, this scale is in line with current studies based on larger-area benthic assessments using reef photomosaic data that are being collected across the tropics for analyses of spatial patterns and structural complexity (e.g. $10 \times 10 \, \text{m}$ plots in [27] and $15 \times 15 \, \text{m}$ plots in [31]) as well as the spatial scale of many active coral outplanting and restoration efforts.

The model is parametrized for Caribbean coral reef benthic community dynamics and includes cells representing four spatially dominant, mutually exclusive functional groups: slowly evolving stony coral (CO), early successional turf algae (TA), later successional macroalgae (MA) and crustose coralline algae (CCA), which act as a type of foundational substrate for growth of the other forms. Ecological processes determining the time evolution of the functional groups include competition, growth, recruitment, algae succession, mortality and herbivory (figure 1a). Model simulations were run with time steps of 0.025 years for durations that included decay to steady-state attractors (10 to approx. 500 years). All rates used are based on the original model [30] and references therein.

Competitive outcomes between functional groups are based on a competitive hierarchy which, in order of decreasing dominance, is large CO (cells belonging to colonies larger than the dominant colony size threshold, $C_{th} = 900 \, \text{cm}^2$), MA, small CO (cells belonging to colonies below the colony size threshold), TA and CCA. The competitive hierarchy, including the existence of a coral colony size threshold above which colonies grow and survive, has been well documented through tracking of coral colonies in regular contact with MA in field observations [32–34]. CCA in the model also represents open space, ignoring possible facilitating or competitive effects of certain CCA species [35]; inclusion of such facilitation or competitive effects has been confirmed to not substantively affect model results [7]. All CO cells were set to represent branching coral types, which exhibit potentially unlimited growth, although dynamics are qualitatively similar for 'massive' coral with colony-specific asymptotic growth [30].

Cell transition probabilities in the model are a function of the current state of a cell and that of its four adjacent neighbour cells (von Neumann neighbourhood), with outcomes determined by the competitive dominance hierarchy. Growth of CO, TA and MA occurs clonally (vegetatively) by laterally expanding into neighbouring space. The probability of an individual cell being overgrown by a neighbouring cell is dependent on the growth rate of the neighbouring functional type, $G_x$, where $x$ denotes the functional type CO, TA or MA, and $n_x$, ranging from 0 to 4, denotes the number of neighbouring cells occupied by the functional type $x$. Therefore, the probability of a cell being overgrown by functional type $x$ is $1 - (1 - G_x)^{n_x}$. Growth rates for CO, TA and MA are 0.01, 1.0 and $0.5 \, \text{m yr}^{-1}$, respectively; previous sensitivity analyses have found model results are robust to changes in growth parameter values [30].

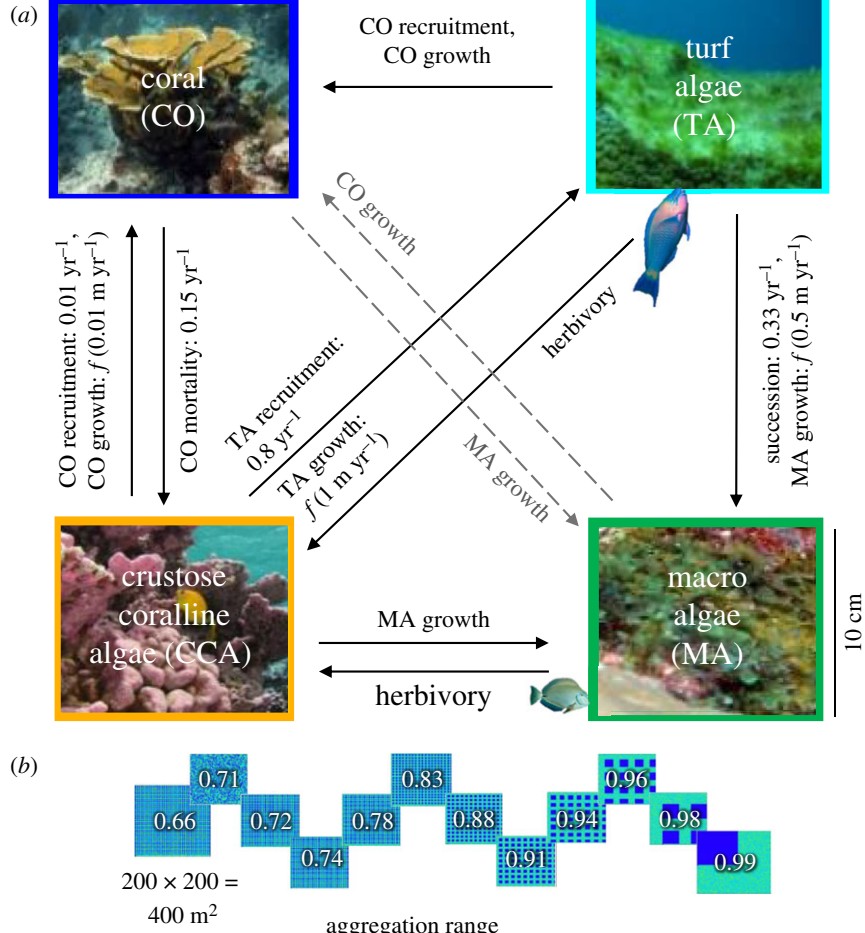

**Figure 1.** (a) Cell transitions among the four functional groups (outlined in representative colour used in lattice) in the model are based on characterizations of coral reef dynamics as transition probabilities or state-dependent functions ($f(x)$) that define probabilities, including competition (dashed arrows), growth, recruitment, algae succession, mortality and herbivory. (b) Model is initialized with various arrangements of coral patches with aggregation varying from 0.66 to 0.99 (fractional cover held constant). (Colours outlining images represent groups in lattice.)

Recruitment of benthic types occurs through the arrival and survival of mobile propagules into the juvenile population present on the reef. In the model, recruitment for each functional group is the composite probability of planktonic arrival in addition to survival into the juvenile class (i.e. growth to the size of a cell on the lattice). CO recruits to cells occupied by either CCA or TA with probability $0.01\ yr^{-1}$; TA recruits to CCA cells with probability $0.80\ yr^{-1}$. Because MA recruits from growing out of TA assemblages on reefs, a TA cell undergoes succession into MA with probability $0.33\ yr^{-1}$. If a cell is designated to be overgrown by a neighbouring cell as well as undergo recruitment, the outcome is determined by the competitive hierarchy.

Natural background mortality can affect all organisms. For algae on the reef, the tendency to exist as an assemblage of multiple individuals limits the likelihood of individual mortality in a particular area. Therefore, in the model, algae mortality occurs only from herbivory (described below). For coral, which are long-lived and dominant competitors, CO cell mortality is simulated with probability $0.15\ yr^{-1}$ and results in CO cells converting into CCA cells.

Herbivory of algae on coral reefs naturally occurs from a suite of reef fishes and invertebrates. Because of their high mobility, herbivorous fish, which are the main herbivores in the Caribbean, can explore wide areas of the benthic reefscape in their search for food. Therefore, grazing is simulated in a spatially random manner across the lattice, with TA and MA cell types having an equal likelihood of being consumed and converted to CCA. Herbivory was set to occur at a moderate grazing rate of 12% of algae cells grazed per time step. This rate is comparable to a median-range mixed assemblage of herbivorous fishes with biomass of approximately $14\ g\ m^{-2}$ [9,36], which decreases at a rate proportional to the total available TA and MA cells if fractional cover of algae decreases on the lattice.

The results presented here are insensitive to the particular ways the ecological processes were parametrized in the model. For example, indirect effects of spatially constrained grazing on coral–algae competition (e.g. from intensified grazing close to large corals that offer fish protection [37,38]) and coupling fish to the benthos, were found to not significantly alter pathway behaviour (refer to electronic supplementary material, ESM8). This does not mean that grazing rate, or herbivory level, has no influence on spatio-temporal dynamics, as has been found elsewhere [9,39] and as demonstrated by our exploration of the impact of levels of herbivory (described at end of this section). Instead, it highlights how the transient stage that arises from more clumped spatial configurations is robust to the grazing algorithm employed. Lastly, algal food preference by grazers previously has been shown to have limited impact on model results [7]. For further details on the parametrization and construction of the coral reef benthic model, refer to Sandin & McNamara [30].

## 2.2. Aggregation

The overall time-evolution of the reefscape was characterized using (i) pathways of fractional cover $f_x$, where $x$ is CO, TA, MA or CCA, calculated as the area occupied by each functional type of cell divided by the total area of the lattice and (ii) coral aggregation ($ag_{co}$), defined in equation (2.1) as the number of coral-to-coral cell boundaries divided by the total number of CO cell boundaries over the lattice (N * M),

$$ag_{co} = \frac{\sum_{i=1}^{N} \sum_{j=1}^{M} (r_{co}(i,j)\, n_{co}(i,j))}{4\, N * M\, f_{co,}} \tag{2.1}$$

where $r_{co}\,(i,j)$ is 1 if the $(i,j)$ cell was occupied by CO and 0 otherwise and $n_{co}(i,j)$ is the number of neighbouring CO cells (0–4). Fractional cover of CO is represented by $f_{co}$. This metric for aggregation is 0 when no two CO cells are neighbours and approaches 1 when all CO cells are arranged in a single clump.

To investigate the effect of aggregation on reefscape pathways, 12 initial configurations varying only in aggregation from 0.66 to 0.99 with CO/TA/MA/CCA initial fractional covers of 0.3/0.5/0.20/0.0 were constructed (figure 1b). These configurations were created using gridded patterns of CO cells with varying distances between colonies of size $30 \times 30$ cm (this size maximized the number of possible configurations using colonies above the competitive size threshold), plus one configuration with random placement of coral colonies (aggregation = 0.71). Each initial configuration was used to initiate 16 simulations, each with a different random number generator seed. All simulations were conducted without disturbances (e.g. storm events) because our focus is on the intrinsic ecological dynamics of the reefscape and because numerical experiments conducted with stochastic disturbance events did not significantly influence the delay effect of aggregation on reefscape pathways (electronic supplementary material, figure S6).

## 2.3. Pathway characterizations

Quantitative features along pathways (e.g. inflection points) yielded four characteristic stages: repelling, transient, attracting and attractor stages (figure 2). For the repelling and attracting stages, the time scale (i.e. the response time related to the internal dynamics of the system) was calculated over the duration of the stage, whereas stage duration was determined for the transient. Boundaries between stages were defined quantitatively using automated algorithms tailored for these simulated time series based on knowledge of the types of solutions possible in the different stages. Specifically, the linearized solutions to escape from and decay to the repellor/attractor in the repelling/attracting stages are exponential rise/decay [40]. Generally, boundaries were determined by following pathways forward in time and flagging the time step where the slope of a fit to an exponential function over a window of three time steps changed by more than 1.5 times the standard deviation of the slope in the previous time step window. For each random-seed-based set of aggregation levels, the boundary between repelling and transient stages first was determined for the initial configuration that arrived at the attractor the fastest (the most regular initial aggregation pathway). To effectively compare the time scales of short-duration repelling stages while minimizing effects of noise from longer-duration initializations, this same repelling-to-transient boundary value then was used for the remaining pathways in that set. The boundary between transient and attracting stages was found by identifying the inflection point (where the derivative of the pathway within the transient stage changes sign). The

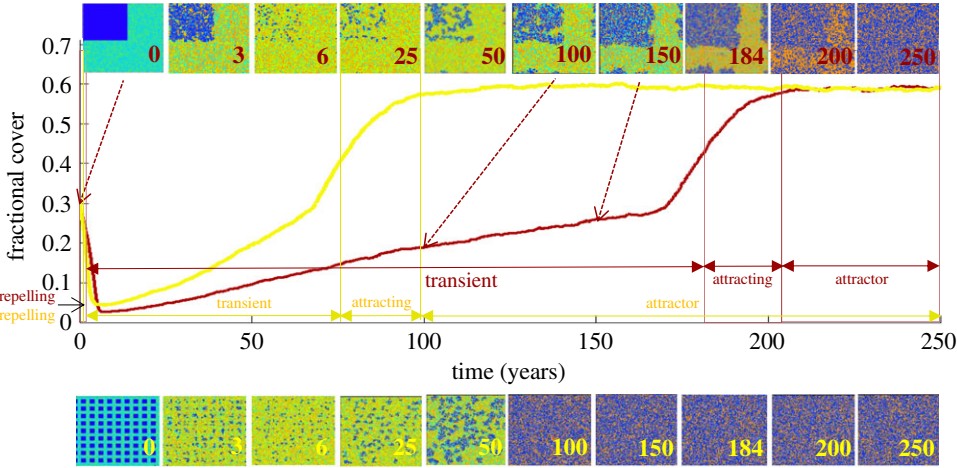

**Figure 2.** Coral fractional cover pathways versus time from initial aggregation level 0.91 (yellow) and 0.99 (dark red) illustrate differences in the four pathway stages: repelling, transient, attracting and attractor. Snapshots of the reefscape lattice at multiple points throughout the pathways (top row for red pathway; bottom row for yellow) show persistence of initial pattern into the attracting stage for both pathways. Blue, CO; green, MA; bluish green, TA; orange, CCA.

boundary between the attracting stage and the attractor stage was found by moving forward along the pathway to flag the point where the slope of a fit to an exponential function over a window of 10 time steps differed by more than two standard deviations of the slope in the previous window. Because the position of this point is sensitive to small stochastic fluctuations arising from probabilistic cell transitions, the fractional cover pathway was smoothed by employing a commonly used iterative diffusion method, in which the diffusion equation is applied to the curve for selectively smoothing at small scales [41].

## 2.4. Herbivory scenarios

The level of herbivory on the reef is expected to impact coral–algae competition, because reefs with depressed herbivore abundances favour growth of competitively dominant algae [42]. Such conditions can influence expectations of spatial configurations with important implications in reef conservation management. In the model, the level of algae herbivory was varied to investigate if pathway dependence on initial aggregation (using four representative aggregation levels: 0.66, 0.77, 0.89, 0.99) changed in herbivore-abundant (24% of algae grazed per time step) versus herbivore-limited (4% of algae grazed per time step) reef conditions ($n = 50$).

## 2.5. Characterizing underlying dynamics (linear versus nonlinear)

To assess the nature of the dynamics underlying the pathways of the reef system, spatio-temporal forecasting was applied to the transient, attracting and attractor stages. The repelling stage is too short for this method. Temporal and spatial forecasting operate by reconstructing state space using lagged replicates of a portion of a series of points sampled in time [43] or space and time [44]. Here, the reconstructed state space consisted of a cube of cells with two of the directions being spatial snapshots of the lattice and the third being time at intervals of one year [45]. The reconstructed state space is then probed to explore the role of nonlinearity by evaluating prediction skill along a given trajectory in the state space as a function of the number of neighbour trajectories used to make the prediction (30 000 points using fivefold cross validation). Forecast accuracy remaining constant with increasing number of nearest neighbours is consistent with linear dynamics. Forecast accuracy that reaches a maximum at moderate number of nearest neighbours and then decreases is consistent with nonlinear dynamics.

## 3. Results

Each of the quantitatively defined stages (figure 2: repelling, transient, attracting, attractor) is related to the underlying modelled ecological dynamics. The repelling stage captures the initial response of the

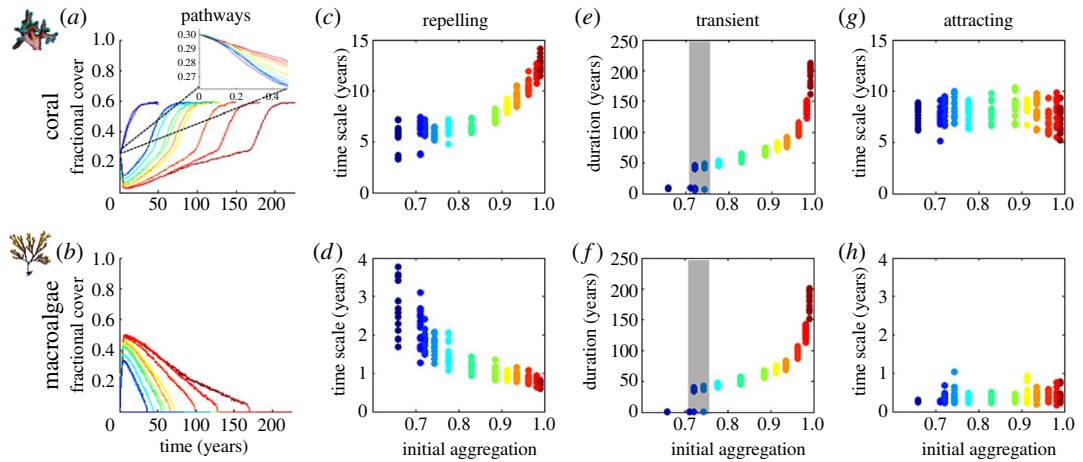

**Figure 3.** Effect of increasing initial aggregation of CO patches on CO and MA fractional cover pathways for 12 different initial aggregation configurations ($n = 16$ per initialization). Colour gradient from blue to red in all panels represents increasing initial aggregation from the most regular (dark blue) to the most clumped (dark red). Temporal pathways of fractional cover for (a) CO (with inset of early starts) and (b) MA: both CO and MA experience an increased delay in arrival to the attractor as initial aggregation of CO increases; time scale of repelling stage as a function of initial aggregation for (c) CO and (d) MA: repelling time scale doubles for CO while the time scale decreases for MA between two initial aggregation extremes; duration of transient stage as a function of initial aggregation for (e) CO and (f) MA: transient duration increases as initial aggregation increases for both, with threshold region (grey) showing simulations with same aggregation realized with and without transient stage; time scale of attracting stage as a function of initial aggregation for (g) CO and (h) MA: attracting time scale unaffected by initial aggregation.

system to the initial aggregation condition. Coral (CO) fractional cover pathways exhibit an initial decline attributed to limitations imposed on CO growth by the limited number of CO cells bordering non-CO cells. During this stage, pathways exponentially move away from a repelling fractional cover value, as predicted by linearization around a repellor. In the transient stage, late successional MA (transitioning from early successional, rapidly recruiting TA cells) actively competes with CO. The transition to the attracting stage is marked by an increase beyond a threshold number of dominant CO cell neighbours of MA, where MA becomes patchy or fragmented enough to expose sufficient borders for the dominant CO to overgrow it and for grazers to consume the remainder. The transition to the attracting stage, where pathways exponentially decay towards the attractor, is marked by steady recruitment and growth of CO onto the foundational substrate—CCA—and TA cells. In the attractor stage, the balance between CO overgrowth of CCA/TA and CO mortality results in a simulated reefscape dominated by CO ($0.586 \pm 0.002$ s.d. fractional cover and $0.640 \pm 0.003$ s.d. aggregation) and the passive CCA.

Increased aggregation in the initial configuration (blue to red pathways in figure 3) affects CO and MA (figure 3a and b, respectively) pathways by prolonging arrival to the attractor. As initial aggregation increases, each CO fractional cover pathway exhibits an increasingly lower minimum (figure 3a). This change reflects CO mortality outweighing CO growth more intensely when CO cells become limited largely by CO cell neighbours that impede growth into surrounding space. CO fractional cover pathways begin to recover from the decline only until enough CO cells undergo mortality, thereby opening space (de-aggregating the pattern) for CO growth onto newly recruited CCA and TA. The CO repelling exponential time scale increases markedly as aggregation increases, with a doubling of the time scale between dispersed and clumped cases (figure 3c). This signifies that aggregation can constrain small-scale ecological processes of growth and competition that manifest in the pathways of fractional cover. The MA repelling time scale (figure 3d) exhibits a trend in the opposite direction, decreasing as aggregation increases in a manner that appears to be coupled to CO. MA competitive losses to the dominant CO cells are minimal and localized to the perimeter of the clumped CO, so that MA more rapidly takes over the inferior TA/CCA dominating the lattice, thereby reducing the MA time scale as aggregation increases. Transient durations for both CO and MA (figure 3e,f) show a sharply increasing trend as aggregation increases, with the difference in transient duration between dispersed and clumped initial conditions exhibiting a 10-fold increase. This is consistent with the duration of the influence of the initial condition found from power spectral

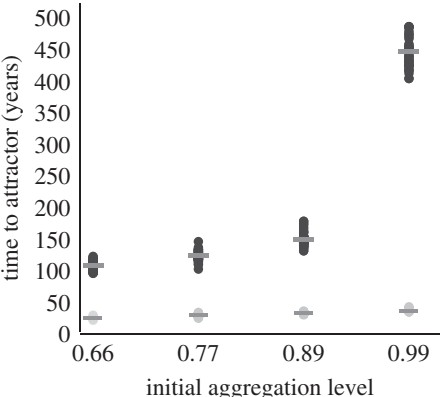

**Figure 4.** Duration of time to arrive at attractor versus initial aggregation for herbivore-abundant (light dots) and herbivore-limited (dark dots) reefs. Degraded herbivore-limited reefs amplify the effect of aggregation on the time duration to arrive at the attractor. Means for each treatment ($n = 50$) are shown as bars.

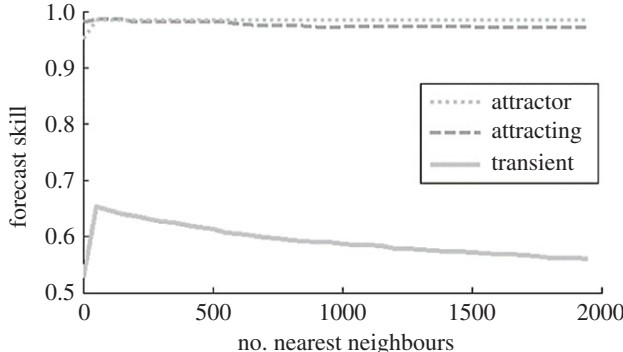

**Figure 5.** Nonlinear spatio-temporal forecasting of a clumped (0.96) initialization pathway tests for nonlinearity for the transient, attracting and attractor stages. Forecast skill versus number of nearest neighbours used for forecasts. Nonlinear dynamics dominates transient stage and linear dynamics dominates attracting and attractor stages. The standard deviation for the attractor stage averaged 0.002 (range: $0.002 - 0.006$), 0.0017 (range: $0.0010 - 0.0039$) for the attracting stage and 0.014 (range: $0.010 - 0.027$) for the transient stage.

analyses (electronic supplementary material, figure S1). The transient stage is mostly absent (duration between 0 and 5 years) for initial aggregations less than 0.72, present in only some iterations for initial aggregation up to 0.75, and rises sharply to greater than 40 years thereafter, suggesting a threshold initial aggregation exists for the presence of a transient stage (figure 3$e$,$f$). Initial aggregation does not influence attracting time scales of either CO or MA (figure 3$g$,$h$) because the initial pattern at this late stage nearly disappears (figure 2 snapshots). In summary, initial spatial configuration is critical for determining the presence and length of the transient stage along reefscape pathways.

A second set of conservation scenario experiments to investigate how varying reef condition affects the results indicates that higher initial aggregation significantly increases the total time required to arrive at the attractor by a factor of approximately 1.5 for herbivore-abundant reefs and approximately 4.0 for herbivore-limited reefs (figure 4; electronic supplementary material, figure S4). For the most clumped initial aggregation (0.99), the time delay in arriving at the attracting stage varies 10-fold between intact herbivore-abundant and degraded herbivore-limited reefs.

To investigate the nature of the dynamics underlying the stages, a spatio-temporal forecasting method was employed [44,45]. The results are consistent with dominance of linear dynamics, or weak one-way interactions (e.g. growth), in the attracting and attractor stages, and nonlinear dynamics, or strong two-way interactions (e.g. active coral–algae competition), in the transient stage (figure 5). Forecast skill is similar for all pathways across aggregation levels, except for cases with short transient stages (less than 5 years), for which the strength of nonlinearity is marginal (electronic supplementary material, figure S3).

In all simulation results described above, attracting and attractor stages are marked by a lack of spatial clumping significantly above the size of a cell. The lack of spatial patterning appears to be a characteristic of

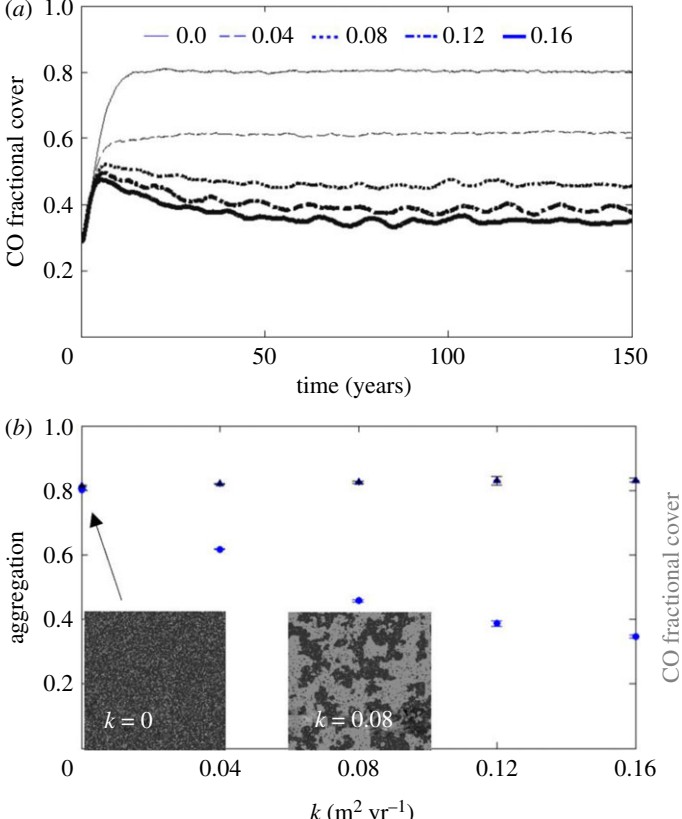

**Figure 6.** Incorporating CO diffusion into the model enables emergence of a long CO fractional cover time scale that leads to self-organization and clumping in the attractor, but also depresses the steady-state CO fractional cover. (*a*) Coral fractional cover pathways for diffusion constant $k = 0$ m$^2$ yr$^{-1}$ (thin) to $k = 0.016$ m$^2$ yr$^{-1}$ (thick) show increasing time scale to arrive at attractor. (*b*) Attractor values for CO fractional cover (circles) and aggregation (triangles) as a function of coral diffusion, $k$. As diffusion increases, CO fractional cover decreases and aggregation increases slightly. Error bars are 1 s.d. and snapshots are of lattice at attractor for $k = 0$ and $0.08$ m$^2$ yr$^{-1}$ (dark, CO; grey, CCA).

the dynamics of this particular model, because the ratio of decay time scales of fast- (i.e. coral growth) and slow-scale (i.e. fractional cover and aggregation) processes is near unity (further details in electronic supplementary material, ESM5). Spatial patterning requires scale separation, or the decoupling of dynamical interactions between different scales (e.g. in forests: time scales of tree growth are significantly shorter and scale separated from time scales associated with development of forest patterns [46]). Scale separation, in turn, requires dissipation, which acts to smooth, damp or mix differences between elements of the system. So dissipation in reefscapes, in the form of diffusion, involves smoothing out of interfaces between patches of coral and other functional groups and could have a biological origin in coral colony fusion processes. Dissipation enforces scale separation by damping fast-scale dynamics on the slow scales of the pattern leading to self-organized spatial configurations [47,48]. To test whether adding dissipation to the model would induce aggregated reefscape patterns in the attractor, the model was modified to include CO diffusion using a range of diffusion constants (figure 6; electronic supplementary material, ESM5). The results indicate that diffusion enables self-organization and clumping in the model, which is in line with previous studies that have linked diffusive or differential flow processes with regular pattern formation [47,49]. However, diffusion also acts to depress the steady-state CO fractional cover. This agrees with the previously quantified effect of increased initial aggregation, where fewer coral borders open to competitive interactions result in mortality processes outweighing growth processes. CO diffusion results in retention of some of that clumpiness, or growth limitation, which translates to a lower CO fractional cover at the attractor.

## 4. Discussion

A dynamical analysis of a simulated coral reefscape shows that pathways starting from clumped initial conditions consistently can be categorized into four stages: repelling (early rapid shift), transient (slow

steady increase from spatial rearranging), attracting (decay to endpoint attractor) and attractor (steady state). Our model results suggest that aggregation sets the context for the processes of coral growth and coral–algae competition. As initial aggregation increases, coral repelling time scale and duration of the transient stage increase because the length of coral borders exposed to non-coral interactions decrease and thereby limit coral expansion (through growth and competition). This model result reflects ecological processes driven by neighbourhood interactions at patch boundaries where, depending on conditions (for reefs: herbivore abundance, nutrient and/or sediment input, etc.), outcomes of competition are determined. Combined with other modelling and experimental studies that highlight the influence of boundaries between competitors [15,19,50], our model results suggest a boundary-centred focus for future reefscape models that build on and go beyond lattice models of reefs. For example, one promising approach involves using continuum models that integrate over both space and time to capture the dynamics of interacting patches of coral reef functional groups.

For reef restoration initiatives that seek to maximize the growth of coral colony fragments cemented to artificial structures of various geometries and arrangements [51], our model results suggest that increasingly aggregated configurations of large, competitively dominant fragments can limit growth in both herbivore-abundant and herbivore-limited systems (figure 4; electronic supplementary material, figure S4). Our results also highlight how smaller, competitively inferior coral colonies can persist in a clumped pattern within the context of a degraded herbivore-limited reef state (electronic supplementary material, figure S8), owing to the reduction of deleterious coral–algae borders. This effect is enhanced if fusion between genetically similar fragments occurs and increases coral survival, given the improved ability of larger individuals to withstand assaults [52,53]. When considering the establishment of protected reef zones for restoration, complementing an evaluation of reefscape local condition with level of coral colony aggregation can help target locations that maximize persistence or coral growth and dominance.

The significance of a protracted transient stage dominated by nonlinear dynamics is that the interactions of organisms are more actively, or strongly, linked during this stage (e.g. in this study, active competition between coral and macroalgae). In other words, competition could appear different during transient (recently disturbed) periods than during steady-state 'pristine' conditions (long after a disturbance). Further studies of reefscapes in transient stages, as when recovering from recent perturbations or when intact, herbivore-abundant reefscapes are on the path towards herbivore-limited, macroalgae-dominated systems could continue to inform conservation monitoring studies and interventions [54].

The existence and duration of transients in the model depends on clumped competitively dominant coral colonies gradually outcompeting algae. The relationship of transients to initial clumping configurations is supported by the existence of a threshold aggregation above which the dynamically nonlinear-dominated transient stage emerges. Previously, transient behaviour has been linked to increasing growth rates in models that follow density-dependent population dynamics as a function of both space and time [55–57]. However, this is the first time, to our knowledge, that a threshold aggregation for transient behaviour has been identified in modelled subtidal systems. These model results are consistent with the argument that spatial distribution, as characterized by aggregation, plays a role in determining the nature of pathways towards large-scale system structure and function. Therefore, level of aggregation complements metrics such as competition coefficients, density and the abundance of competitors in characterizing systems dominated by intense competition for space, although investigating explicit links to non-coral systems (e.g. tidal marshes, arid systems, etc.) will require futher research [19,47].

The role of aggregation as an additional key spatial metric of reefscape dynamics also is evident when constructing the reefscape state space (i.e. the multi-dimensional space of a dynamical system where all possible states can be mapped). Previous studies of reefscape dynamics have used only two dimensions, coral and macroalgae fractional cover, as the axes of state space (e.g. [9,30]). However, in the reefscape model, trajectories in a two-dimensional state space of coral and macroalgae fractional cover intersect themselves and other trajectories when varying initial aggregation (figure 7a), indicating that two dimensions (coral and macroalgae fractional cover) do not adequately represent the dynamics of the reefscape model (because trajectories that resolve dynamics do not cross in state space [58]). Adding aggregation as a third dimension of state space removes most trajectory crossings and further resolves trajectory pathways towards the attractor; a finding that also is supported by an independent method (method of false nearest neighbours: the fraction of false nearest neighbours drops to near zero at three embedding dimensions) [59] (figure 7b; electronic supplementary material, figure S2). That benthic coral reef system dynamics are better represented using three dimensions supports our earlier results on the key role of aggregation in spatio-temporal dynamics. This theoretical finding indicates that comparatively large-scale reefscape field observations that can track aggregation to generate long-term

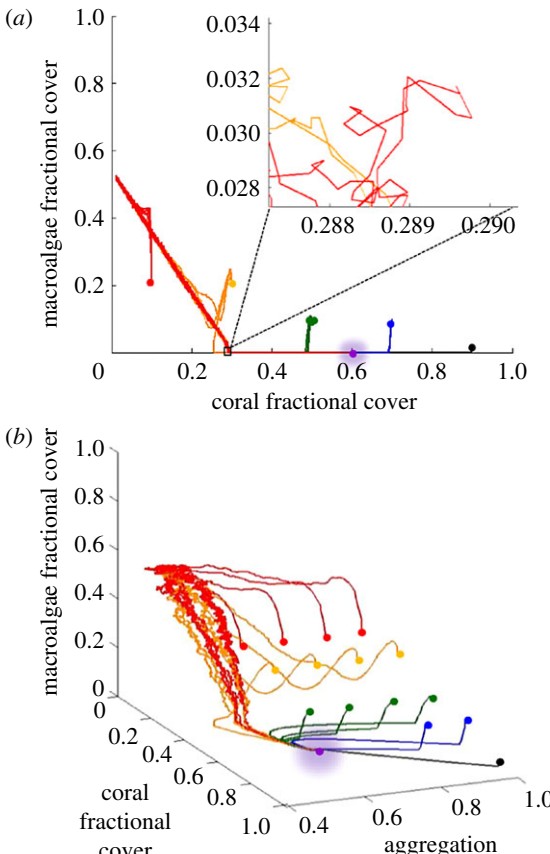

**Figure 7.** Simulated reefscape state space in two dimensions (*a*) and three dimensions (*b*), illustrates how addition of aggregation axis removes trajectory intersections and resolves dynamics showing the point attractor at C0 fractional cover $0.586 \pm 0.002$ s.d., macroalgae fractional cover less than 0.001 and aggregation $0.640 \pm 0.003$ s.d. Trajectories colour coded in groups of initial coral fractional cover, with dots indicating initial state and glowing purple dot the attractor for all trajectories. Initial C0 fractional covers: 0.10 (red), 0.30 (gold), 0.50 (green), 0.70 (blue) and 0.90 (black); initial aggregation: 0.66, 0.77, 0.89 and 0.99.

spatio-temporal series (as recently has become possible with large-area ortho-rectified two- and three-dimensional photomosaics that resolve the boundaries of coral [27,60]) might contribute towards better resolving reefscape dynamics.

Ecological modelling and experimentation typically employ random initial configurations [61,62]; our model results suggest that clumped initial configurations can significantly influence dynamical pathways in spatially extensive systems, and so measuring aggregation can inform analyses, in our case, of coral reef dynamics. In coral reefs, these theoretical results can be tested in long-term monitoring or restoration experiments covering large-scale segments of reefs (approx. 100 m$^2$ or more with biannual or annual surveys) by tracking coral colonies out-planted over a range of aggregation levels. We are aware of at least one experiment that has been initiated based on our results (J.E.S. 2017, personal communication). Generally, the resulting delay effect arising from more clumped initial configurations on dynamical pathways suggests that analyses of models of other spatially extended systems might benefit from considering the effect of initial spatial configurations that deviate from random distributions on modelled system dynamics.

Under rapidly changing environmental conditions owing to global warming and other anthropogenic impacts, coral reefs can be increasingly perturbed far from their steady state [63,64]. A brief model exploration of recovery from storm disturbances showed decreased persistence of coral on reefscapes with pathways retaining initial clumped configurations characteristic of the transient stage (refer to electronic supplementary material, ESM6; figure S7). Future work could further investigate the impact of recurrent and intensified storm disturbances on transient dynamics and underlying spatial configurations, although broad generalizations might be challenging owing to the site-specific nature of disturbances [65].

Coral reefs are complex systems exhibiting variations across basins; these variations provide a multitude of opportunities for testing and expanding the implications of our theoretical research [26,66]. For example, macroalgae interactions and algal colonization have been observed to be far less frequent in the Indo-Pacific than in the Caribbean [67], which could affect the relative durations of transient dynamics in the two

regions. Studying locations with a larger number of benthic groups, with differing life-history traits and growth morphologies, could also illuminate aspects of transient dynamics, especially because an overall tendency for clumping has been observed across taxa [26–28,68]. Another possible study could test how aggregation affects reefscape pathways in conditions dominated by intransitive competitive interactions, or indirect interactions between multiple players, rather than transitive, or direct interactions (which has been our focus here) [69].

In conclusion, initial spatial clumpiness, as characterized by aggregation, delays arrival of simulated coral reef fractional cover pathways towards their endpoint attractors by modulating the duration of the transient stage (one of the four characteristic dynamical stages defined). This delay effect on pathways is magnified (10-fold) in herbivore-limited reef conditions, where coral growth is further slowed by conditions that favour algae competitors. The character of the dynamics at each stage of a reefscape pathway is determined by aspects of the underlying ecology; dynamics in the transient stage are dominated by nonlinear, competitive interactions, whereas the dynamics of the other stages are dominated by linear interactions, such as growth. Coral fusion, simulated using diffusion, promotes pattern formation in the model, which suggests that dissipative dynamics, in addition to nonlinear dynamics, should be considered in future research on coral reef models. Three variables, coral and macroalgae fractional cover and aggregation, are required to resolve state space dynamics of clumped coral reefscapes, underlining the key role of aggregation in these substrate-bound systems.

Data accessibility. Code for Pathway Boundary Detections: https://github.com/2mangitos/boundarydivs2.git. Code for nonlinear forecasting: https://github.com/NickC1/skedm. Dataset for 'aggregation effect on time scale and stage durations' has been uploaded as part of the supplementary material.

Authors' contributions. M.B-M. conceived of study, participated in design and execution of modelling and analysis, drafted the manuscript; B.T.W. participated in design and execution of modelling and analysis, drafted the manuscript; S.A.S. participated in design and helped draft the manuscript; D.E.M. participated in execution of modelling and helped draft the manuscript. All authors gave final approval for publication.

Competing interests. We have no competing interests.

Funding. M.B-M. funded by NSF Graduate Fellowship, the Department of Scripps Institution of Oceanography and The Gordon and Betty Moore Foundation. S.A.S. funded by the Gordon and Betty Moore Foundation. D.E.M. funded by NOAA CIOERT Award NA14OAR4320260. B.T.W. received no funding for this project.

Acknowledgements. Insightful conversations with and technical support from N. Cortale and Y. Eynaud are gratefully acknowledged. We are also grateful to the anonymous reviewers whose comments improved the manuscript. This work was supported by an NSF Graduate Fellowship to M. Brito-Millán, the Department of Scripps Institution of Oceanography and The Gordon and Betty Moore Foundation. Author order ranked by contribution.

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
