## [Reviewer comments · Royal Society Open Science]

Review History

Decision letter (RSOS-181551.R0)

05-Oct-2018

Dear Dr BRITO-MILLAN:

Manuscript ID RSOS-181551 entitled "Influence of Aggregation on Benthic Coral Reef Spatio-temporal Dynamics" which you submitted to Royal Society Open Science, has been reviewed. The comments from reviewers are included at the bottom of this letter.

In view of the criticisms of the reviewers, the manuscript has been rejected in its current form. However, a new manuscript may be submitted which takes into consideration these comments.

Please note that resubmitting your manuscript does not guarantee eventual acceptance, and that your resubmission will be subject to peer review before a decision is made.

Your resubmitted manuscript should be submitted by 04-Apr-2019. If you are unable to submit by this date please contact the Editorial Office.

Please note that Royal Society Open Science will introduce article processing charges for all new submissions received from 1 January 2018. Charges will also apply to papers transferred to Royal Society Open Science from other Royal Society Publishing journals, as well as papers submitted as part of our collaboration with the Royal Society of Chemistry (<http://rsos.royalsocietypublishing.org/chemistry>). If your manuscript is submitted and accepted for publication after 1 Jan 2018, you will be asked to pay the article processing charge, unless you request a waiver and this is approved by Royal Society Publishing. You can find out more about the charges at <http://rsos.royalsocietypublishing.org/page/charges>. Should you have any queries, please contact openscience@royalsociety.org.

on behalf of Professor Anotida Madzvamuse (Associate Editor) and Professor Kevin Padian (Subject Editor)
openscience@royalsociety.org

Subject Editor Comment to Author:

Please note that the rejection is not related to scientific content, which was not reviewed; we just need a clean copy. Thanks.

Associate Editor Comments to Author (Professor Anotida Madzvamuse):

Dear Authors,

This manuscript was submitted to the Journal of the Royal Society Interface and has been transferred. This is all very well, but the submitted manuscript is not in a state to be peer-reviewed since it has comments and annotations in the the pdf. I would recommend that you resubmit a new clean copy that can be used for proper peer review. The resubmission must be treated as a new submission.

Thanks for submitting your manuscript to the Royal Society Open Science.

With best wishes,
Anotida

Author's Response to Decision Letter for (RSOS-181551.R0)

See Appendix A.

RSOS-181703.R0

Review form: Reviewer 1

Is the manuscript scientifically sound in its present form?

Yes

Are the interpretations and conclusions justified by the results?

No

Is the language acceptable?

Yes

Is it clear how to access all supporting data?

Yes

Do you have any ethical concerns with this paper?

No

Have you any concerns about statistical analyses in this paper?

No

Recommendation?

Reject

Comments to the Author(s)

The Authors tackled an interesting problem of spatio-temporal evolution of coral reefs in mesoscopic scale (20*20 m), diving the 2D system to small, 0.1*0.1 m cells, using cyclic boundary conditions. The calculations are correct, conclusions are well-established (it is a transferred manuscript, already revised, at least once). I have only one problem with the results presented in the manuscript:

- although meso-scales are very important, to have good description for the spatial characteristic of a coral reef, it would be very much advised to go above 20*20 meters. It would be interesting to see the validity of the conclusions on spatial scales covering the whole reef. In the present form, the manuscripts present interesting results, but might not be good enough for a leading scientific journal.

Review form: Reviewer 2

Is the manuscript scientifically sound in its present form?

Yes

Are the interpretations and conclusions justified by the results?

Yes

Is the language acceptable?

Yes

Is it clear how to access all supporting data?

Yes

Do you have any ethical concerns with this paper?

No

Have you any concerns about statistical analyses in this paper?

No

Recommendation?

Accept as is

Comments to the Author(s)

This manuscript is a slightly modified version of the one submitted before to Royal Society Interface which I reviewed before. As said in the previous rounds I think this work is written well and since the first version the authors did substantial work to make the structure of the manuscript much more clear. Furthermore they modified some additional points to improve based on additional suggestions the reviewers did. Overall these modifications helped improving the manuscript and therefore I think the publications of the work should not be delayed too much, as the topic is timely and very relevant to the field: (Initial) spatial structure will determine the pace of the transient towards the stable climax state. This kind of insights could for example be very relevant when designing restoration schemes.

There remains a minor point that cannot (and doesn't have to) be resolved (dependent on the editor's opinion):

The authors and I have a fundamentally different view on what can be generalised based on the modelling. The authors feel that their model is too specific, preventing them to discuss their result in context of other spatially structures ecosystems. I do agree that making generalisations should be done with care and caution. Yet, using such a generic tool as an CA... The probabilistic dynamics in such a model are very generic and analogous to many other systems (see for instance Pascual & Guichard 2005 TREE or Kefi et al 2011 in Ecology Letters). Hence, I would at least speculate a bit on the possible larger applicability of your results to other spatially extended systems because in many modelling studies 'typically random initial configurations' (e.g. in L429-435) are used.

But as said, I don't think this difference in insight should preclude a further delay of this interesting modelling study.

Decision letter (RSOS-181703.R0)

04-Jan-2019

Dear Dr BRITO-MILLAN

On behalf of the Editor, I am pleased to inform you that your Manuscript RSOS-181703 entitled "Influence of Aggregation on Benthic Coral Reef Spatio-temporal Dynamics" has been accepted for publication in Royal Society Open Science subject to minor revision in accordance with the referee suggestions. Please find the referees' comments at the end of this email.

The reviewers and Subject Editor have recommended publication, but also suggest some minor revisions to your manuscript. Therefore, I invite you to respond to the comments and revise your manuscript.

Please note that the email address btwerner@ucsd.edu is not currently functional through ScholarOne: you should ensure that Dr Werner adds an additional email address to their account before submitting the revision.

- Ethics statement

- Data accessibility

If you wish to submit your supporting data or code to Dryad (<http://datadryad.org/>), or modify your current submission to dryad, please use the following link:
<http://datadryad.org/submit?journalID=RSOS&manu=RSOS-181703>

- Competing interests

- Authors' contributions

- Acknowledgements

- Funding statement

Because the schedule for publication is very tight, it is a condition of publication that you submit the revised version of your manuscript before 13-Jan-2019. Please note that the revision deadline will expire at 00.00am on this date. If you do not think you will be able to meet this date please let me know immediately.

on behalf of Professor Anotida Madzvamuse (Associate Editor) and Kevin Padian (Subject Editor)
openscience@royalsociety.org

Associate Editor Comments to Author (Professor Anotida Madzvamuse):

I recommend that the authors address fully the concerns of Reviewer 1 in detail and consider this as a major revision. If this can be addressed, then the manuscript could be possibly accepted but in its present form, it is not appropriate.

Editor comments:

Looking at the manuscript, the reviews, and the authors' response to previous reviews, it appears that the authors have attempted to clarify their manuscript and reply to concerns; but the referees still seem to raise the same concerns. The work appears fundamentally sound, and I think the reviewers may simply be asking for a different study than the authors have produced. I would like to accept this with minor revision, asking the authors once again to see if they can clarify any points raised, given that they may anticipate criticisms like these once the manuscript is published. Thanks for submitting. Oh, and before you make any revisions, could you please "un-right justify" the manuscript text? It makes reading and editing easier. Thanks.

Reviewer comments to Author:

Reviewer: 1

Comments to the Author(s)

The Authors tackled an interesting problem of spatio-temporal evolution of coral reefs in mesoscopic scale (20*20 m), diving the 2D system to small, 0.1*0.1 m cells, using cyclic boundary conditions. The calculations are correct, conclusions are well-established (it is a transferred manuscript, already revised, at least once). I have only one problem with the results presented in the manuscript:

- although meso-scales are very important, to have good description for the spatial characteristic of a coral reef, it would be very much advised to go above 20*20 meters. It would be interesting to see the validity of the conclusions on spatial scales covering the whole reef. In the present form, the manuscripts present interesting results, but might not be good enough for a leading scientific journal.

Reviewer: 2

Comments to the Author(s)

This manuscript is a slightly modified version of the one submitted before to Royal Society Interface which I reviewed before. As said in the previous rounds I think this work is written well and since the first version the authors did substantial work to make the structure of the manuscript much more clear. Furthermore they modified some additional points to improve based on additional suggestions the reviewers did. Overall these modifications helped improving the manuscript and therefore I think the publications of the work should not be delayed too much, as the topic is timely and very relevant to the field: (Initial) spatial structure will determine the pace of the transient towards the stable climax state. This kind of insights could for example be very relevant when designing restoration schemes.

There remains a minor point that cannot (and doesn't have to) be resolved (dependent on the editor's opinion):

The authors and I have a fundamentally different view on what can be generalised based on the modelling. The authors feel that their model is too specific, preventing them to discuss their result in context of other spatially structures ecosystems. I do agree that making generalisations should be done with care and caution. Yet, using such a generic tool as an CA... The probabilistic dynamics in such a model are very generic and analogous to many other systems (see for instance Pascual & Guichard 2005 TREE or Kefi et al 2011 in Ecology Letters). Hence, I would at least speculate a bit on the possible larger applicability of your results to other spatially extended systems because in many modelling studies 'typically random initial configurations' (e.g. in L429-435) are used.

But as said, I don't think this difference in insight should preclude a further delay of this interesting modelling study.

Author's Response to Decision Letter for (RSOS-181703.R0)

See Appendix B.

Decision letter (RSOS-181703.R1)

15-Jan-2019

Dear Dr BRITO-MILLAN,

I am pleased to inform you that your manuscript entitled "Influence of Aggregation on Benthic Coral Reef Spatio-temporal Dynamics" is now accepted for publication in Royal Society Open Science.

on behalf of Professor Anotida Madzvamuse (Associate Editor) and Kevin Padian (Subject Editor)
openscience@royalsociety.org

Appendix A

Transfer Submission to Royal Society Open Science

Journal of the Royal Society Interface – 2nd set of edits

Reviewer(s)' Comments to Author:

Referee: 3

Comments to the Author

Referee's report

Revised version; Influence of Aggregation on Benthic Coral Reef Spatio-temporal Dynamics by Marlene Brito-Millan, BT Werner, Stuart A Sandin, Dylan E McNamara

The Authors clarified most of the questions raised by the other referees and in some sense, the manuscript is improved. But I still do believe, that it is not sufficient for publication in the Journal of the Royal Society Interface. The Author reasoned, that the system size used by them is generally accepted within the community; it might be true, but my remark is still valid, that much better resolutions would be required to provide new results. Present supercomputers – and even high-speed PC-s – are able to reach much bigger resolution, than this 200*200, 2m² one.

We feel an important clarification must be made here. The reef area we are simulating IS NOT 2 m². It is a 400 m² (20m x 20m) reef area, a spatial scale large enough to capture cross-reef structural and compositional patterns. This is now explicitly stated from Line 150.

Additionally, using parameters established previously by other low-resolution simulation is not necessarily the way to reach novel results.

From a complex systems perspective, it is widely theorized that small-scale interactions at a constituent level (in our case the scale of the coral colony animal) can lead to emergent patterns at larger landscape levels through processes of self-organization (e.g., Liu et al 2014; Rietkerk & van de Koppel 2008, Kessler & Werner 2003). Model behavior and dynamics have been shown to be robust to the parameters used in our study (as stated in L180).

I still believe that the problem is interesting, the methodology is good, but the calculation and the numerical results are insufficient, therefore I do not recommend the publication.

New Cited References:

- **Q.-X. Liu, P. M. Herman, W. M. Mooij, J. Huisman, M. Scheffer, H. Olf, J. van de Koppel. *Pattern formation at multiple spatial scales drives the resilience of mussel bed ecosystems. Nature Communications 5, 5234 (2014).***
- **Kessler, M.A. and Werner, B.T. (2003) *Self-organization of sorted patterned ground. Science 299, 380–383***

Referee: 2

Comments to the Author

Overall many things have been improved. I think the study and its results are of interest to the readers of Interface, although I still think that the study is still too much focused on coral reef community and the readers of this journal are interested in a broader scope I imagine.

We thank the reviewer for their comment. We have added additional text referring to non-coral systems to further widen the scope. L90 cites work on marsh tussocks; L355 links to papers referring to pattern formation in non-coral ecosystems. Importantly, though, we believe that further analogies and applicability to other ecosystems will depend upon having thorough understanding of the dynamics of the systems in question. We have aimed to be transparent regarding the core dynamics and interactions within our system. We have guesses of analogous systems, but believe that assertions regarding direct analogs is dangerous in terms of scholarship, and believe that it is beyond the scope of this manuscript to assert deep dynamical knowledge of other systems.

The authors solved one of my main concerns by building up better to the final figure (previously Fig6 and now Fig7). They do this e.g. by providing a set of questions (lines 135-144) they investigate before drawing the final conclusions. I like this very much as it provides much more structure to the whole manuscript. This lack of structure in the previous version might have led to me thinking they were hiding the most interesting results in the back of the manuscript. Furthermore, I very much like the additional Figure 1. This makes the possible transitions in the model much more clear.

However, I think it would be good if Fig2 is a bit expanded by showing another transient as well. Now the sequence of spatial arrangements from initial (0) to final (at 250) is shown. But One of the main points of the author is that differences in transient are due to differences in initial spatial configurations. It would be nice if you could see at least two of these differences transients including a sequence of spatial configurations to illustrate this point here.

We have added the additional pathway into figure 2 with its corresponding sequence of spatial configurations. Fig 2 text has been updated to describe changes (L619).

Moreover, I still have some difficulty following the rational behind the demarkations of the transients. The authors claim they have good knowledge of the start and end points and not about the transients. But to me it doesn't become clear how they use this knowledge and how this is applied in the choices for demarcating the different development stages. For instance on what ground do they use a fit to the exponential to part of the transient? I think it could help explaining the rational behind these choices better.

To address the reviewer's confusion about why exponential fits formed the basis of our determination of the transitions between repelling and transient and transient and attracting stages, we added the following sentence to the manuscript starting on L245: "*Specifically, the linearized solutions to escape from and decay to the repellor/attractor in the repelling/attracting stages are exponential rise/decay (Ott, 1983).*"

Additional remarks:

Line 139. Switch order of the nonlinear and linear
Change made (now L139).

Line 154. Why are cylindrical and not cyclic boundary conditions used?

We use periodic boundary conditions in both horizontal directions, in other words, a toroidal lattice, to simulate a domain that is large enough for reefscape patterns to emerge. (now stated in L150)

Appendix B

Royal Society Open Science – Response to Referees (author responses in bold)

Associate Editor Comments to Author (Professor Anotida Madzvamuse):

I recommend that the authors address fully the concerns of Reviewer 1 in detail and consider this as a major revision. If this can be addressed, then the manuscript could be possibly accepted but in its present form, it is not appropriate.

Editor comments:

Looking at the manuscript, the reviews, and the authors' response to previous reviews, it appears that the authors have attempted to clarify their manuscript and reply to concerns; but the referees still seem to raise the same concerns. The work appears fundamentally sound, and I think the reviewers may simply be asking for a different study than the authors have produced. I would like to accept this with minor revision, asking the authors once again to see if they can clarify any points raised, given that they may anticipate criticisms like these once the manuscript is published. Thanks for submitting. Oh, and before you make any revisions, could you please "un-right justify" the manuscript text? It makes reading and editing easier. Thanks.

We thank the editor for their interest in our study. Further, we appreciate the enthusiasm and curiosity that our study appears to have invoked. Modeling studies are constructed with model structures and assumptions that address particular suites of questions. As the Editor identifies, we chose the structure of this model (linked with the definition of the domain) with the goal of answering questions about how clumping at the colony / patch scale (order 10-100m² clumps) influence the rate of dynamical evolution. If we had been interested in rates of dynamical evolution for coastlines to islands (order 1000-10000m² features), a different model structure would have been introduced. Of course, the answers that could emerge from such a study hold immense promise, and we appreciate the enthusiasm of the Associate Editor in highlighting this potential.

In the manuscript, we have sought to include more justification of our goals and approach, which hopefully will provide more context for the reader.

The manuscript has been left-justified.

Reviewer comments to Author:

Reviewer: 1

Comments to the Author(s)

The Authors tackled an interesting problem of spatio-temporal evolution of coral reefs in mesoscopic scale (20*20 m), diving the 2D system to small, 0.1*0.1 m cells, using cyclic boundary conditions. The calculations are correct, conclusions are well-established (it is a transferred manuscript, already revised, at least once). I have only one problem with the results presented in the manuscript:

- although meso-scales are very important, to have good description for the spatial characteristic of a coral reef, it would be very much advised to go above 20*20 meters. It would be interesting to see the validity of the conclusions on spatial scales covering the whole reef. In the present form, the manuscripts present interesting results, but might not be good enough for a leading scientific journal.

We thank the reviewer for their insight. The intermediate spatial scale simulated (on the order of tens of meters) is large enough to capture emergent spatio-temporal patterns based on constituent-scale dynamics (thousands of coral colonies), which was the primary aim of this work. Additionally, this scale is in line with current studies based on reef photomosaic data being collected across the

tropics for analyses of spatial patterns and structural complexity (e.g., 10 m x 10 m plots in Edwards et. al. 2017, 15 m x 15 m plots in Gonzalez-Rivero et. al. 2017, etc.) as well as the spatial scale of many active coral outplanting and restoration efforts.

We recognize the need within the field of coral reef conservation to move towards simulating larger reef areas grounded in the dynamics of the reefscape. This need partially motivated current work that upscaled this paper's lattice model into a PDE model that simulates larger areas (on order of hundreds of meters) based on constituent (coral colony) benthic dynamics (manuscript on this topic in preparation). Importantly, the focal results from this study (i.e., that clumped patterns demonstrate slower dynamics than random patterns) are revealed similarly from our generalized methodology, although with less resolution of the dynamical structure. As such, and despite analytical approaches to search for bias of results, we have found no evidence that the conclusions presented in this manuscript are sensitive to the spatial scale of analysis (specifically, not when considering the effects of colony / patch scale clumping on emergent dynamics).

Finally, the ultimate challenge for conducting very-large-scale simulations will remain identifying comparisons of model results to large-scale time-series data of community structure, as such large-scale data has only recently begun to be collected, digitized and analyzed using archival photographic techniques (with ecological post-processing being the most time-intensive step).

We have included more about the chosen scale based on the above points in the main text from L151:

“The model domain consists of a 200 x 200 cell lattice simulating a 400 m² area with periodic boundary conditions in both horizontal directions (i.e., a toroidal lattice); large enough to capture cross-reef spatial patterns based on constituent-scale dynamics (for thousands of coral colonies). Additionally, this scale is in line with current studies based on larger-area benthic assessments using reef photomosaic data that are being collected across the tropics for analyses of spatial patterns and structural complexity (e.g., 10 m x 10 m plots in (27) and 15 m x 15 m plots in (31)) as well as the spatial scale of many active coral outplanting and restoration efforts.”

References Cited:

27. Edwards C, et al. (2017) Large-area imaging reveals biologically-driven non-random spatial patterns of corals at a remote reef. *Coral Reefs*.

31. Gonzalez-Rivero, M.; Harborne, A.R.; Herrera-Reveles, A.; Bozec, Y.M.; Rogers, A.; Friedman, A.; Ganase, A.; Hoegh-Guldberg, O. Linking fishes to multiple metrics of coral reef structural complexity using three-dimensional technology. *Sci. Rep.* 2017, 7, 13965.

Reviewer: 2

Comments to the Author(s)

This manuscript is a slightly modified version of the one submitted before to Royal Society Interface which I reviewed before. As said in the previous rounds I think this work is written well and since the first version the authors did substantial work to make the structure of the manuscript much more clear. Furthermore they modified some additional points to improve based on additional suggestions the reviewers did. Overall these modifications helped improving the manuscript and therefore I think the publications of the work should not be delayed too much, as the topic is timely and very relevant to the field: (Initial) spatial structure will determine the pace of the transient towards the stable climax state. This kind of insights could for example be very relevant when designing restoration schemes.

There remains a minor point that cannot (and doesn't have to) be resolved (dependent on the editor's opinion):

The authors and I have a fundamentally different view on what can be generalised based on the modelling.

The authors feel that their model is too specific, preventing them to discuss their result in context of other spatially structures ecosystems. I do agree that making generalisations should be done with care and caution. Yet, using such a generic tool as an CA... The probabilistic dynamics in such a model are very generic and analogous to many other systems (see for instance Pascual & Guichard 2005 TREE or Kefi et al 2011 in Ecology Letters). Hence, I would at least speculate a bit on the possible larger applicability of your results to other spatially extended systems because in many modelling studies 'typically random initial configurations' (e.g. in L429-435) are used.

But as said, I don't think this difference in insight should preclude a further delay of this interesting modelling study.

We thank the reviewer for their insight. We have added the following sentence to emphasize the need to consider the effect of non-random initial configurations in other spatially extended systems, although such considerations are beyond the scope of this study (from L447):

“Generally, the resulting delay effect arising from more clumped initial configurations on dynamical pathways suggests that analyses of models of other spatially extended systems might benefit from considering the effect of initial spatial configurations that deviate from random distributions on modeled system dynamics”.